# Characterization of postsynaptic glutamate transporter functionality in the zebrafish retinal first synapse across different wavelengths

Marco Garbelli, Stephanie Niklaus, Stephan CF Neuhauss*

University of Zurich, Department of Molecular Life Sciences, Zurich, Switzerland

## eLife Assessment

This **important** study reveals that Excitatory Amino Acid Transporters play a role in chromatic information processing in the retina. The combination of (double) mutants, behavioral assays, immunohistochemistry, and electroretinograms provides **solid** evidence supporting the appropriately conservative conclusions. The work will be of interest to neurobiologists working on color vision or retinal processing.

*For correspondence:
stephan.neuhauss@mls.uzh.ch

Competing interest: The authors declare that no competing interests exist.

**Abstract** In the zebrafish retina, incident light undergoes wavelength-dependent processing encompassing mechanisms such as color opponency, contrast enhancement, and motion detection prior to neural transmission to the brain proper. In darkness, photoreceptors continuously release glutamate into the synaptic cleft, a process that diminishes in response to increased light intensity, thereby conveying visual signals to ON and OFF bipolar cells. Specifically, in zebrafish, the ON pathway signal transduction is mediated by metabotropic glutamate receptor 6b (mGluR6b) and Excitatory Amino Acid Transporters (EAATs). Here, we demonstrate that the knockout of *eaat5b* and *eaat7* disrupts electroretinogram responses to short and long-wavelength stimuli while preserving middle-wavelength responses, suggesting wavelength-specific roles. We found differential expression of EAAT5b and EAAT7 in the outer plexiform layer, particularly in the strike zone, crucial for prey capture, supporting task-specific involvement of these signaling pathways. In order to investigate this, we developed a virtual hunting assay using UV light stimuli. Such a behavioral assay targeting short and long wavelengths indicates that EAAT5b and EAAT7 influence UV-dependent prey detection and motion sensing differently. Our findings highlight the importance of EAAT5b and EAAT7 in modulating light integration dynamics in the zebrafish retina.

## Introduction

The developing zebrafish retina represents an exquisite example of functional adaptation and optimization to the visual environment these fish inhabit. Zebrafish are found in the Indian subcontinent, living in shallow ponds and still water bodies such as rice fields (*Spence et al., 2008*). During monsoon season, they can be found in streams and rivers with a low flow regime where these animals mostly occupy its margins (*Spence et al., 2008*). Being diurnal animals with a cone-dominant retina, the adult photoreceptor layer features a uniform mosaic structure where the different photoreceptor subtypes are homogeneously distributed over the whole surface (*Allison et al., 2010*). In the wild, zebrafish larvae are exposed to the full spectrum of sunlight passing through the shallow waters in which they reside. Compared to the crystalline organization found in the adult photoreceptor layer, the larval eye

displays marked anisotropies in the distribution of cell types in both outer and inner retina, possibly to capitalize on the environmental lightning (*Zimmermann et al., 2018*). Photoreceptor subtypes and their underlying chromatic and achromatic networks present higher densities in specific regions that match with the distributions of wavelengths found in its natural environment (*Zimmermann et al., 2018*). Dissection of the zebrafish visual circuits organization revealed the existence of at least three non-opponent achromatic axes (achromatic-OFF, UV-ON, Red ON-OFF) and an equivalent number of opponent chromatic ones (UV/RGB, R/G, RG/B) (*Orger and Baier, 2005*; *Zimmermann et al., 2018*; *Yoshimatsu et al., 2020*; *Bartel et al., 2021*). Amongst the achromatic ones, the UV-ON and the Red ON-OFF axes appear to be directly involved (if not essential) in specific behaviors such as prey-capture for the former (*Yoshimatsu et al., 2020*) and motion sensing for the latter (*Orger and Baier, 2005*). This suggests that the structure of the larval retina is adapted to maximize the extraction of environmental information, thereby enhancing the accuracy and sensitivity of behaviors essential for vital functions such as feeding and navigation. However, the internal processing within the central nervous system that regulates these circuits remains less understood. In zebrafish larvae, the majority of visual pathways are established at the first synapse (*Yoshimatsu et al., 2021*). Here, the wavelength components of incoming light are separated by the four cone subtypes. Distinct horizontal cells (HCs) groups subsequently contribute to the generation of the first opponent axes (R/G, RG/B) (*Yoshimatsu et al., 2021*). Bipolar cells (BCs) receive these signals and transmit it to ganglion cells (GCs) with further modulation from amacrine cells (ACs). Finally, GCs will send the processed information directly to the brain proper. It was recently demonstrated that the inner retina is crucial for establishing two of the known visual axes (achromatic-OFF, UV/RGB) and for the introduction of a temporal component to the response (transient or sustained) (*Bartel et al., 2021*; *Wang et al., 2023*). However, most of the chromatic processing is computed at the first retinal synapse and is conserved throughout the vertical pathway. In darkness, photoreceptors are depolarized and tonically release glutamate in the synaptic cleft. This causes OFF-bipolar cells that express glutamate ionotropic receptors to depolarize upon light decrease (*Euler et al., 2014*). Conversely, ON-bipolar cells express a metabotropic glutamate receptor (mGluR6), which causes the blocking of TRPM1, a constitutively open cation channel (*Morgans et al., 2009*; *Morgans et al., 2010*). When light strikes the eye, the photoreceptors hyperpolarize and gradually reduce glutamate release. This leads to a reopening of TRPM1 and depolarization of the ON bipolar cell (*Morgans et al., 2009*). Electrophysiological experiments confirmed that in zebrafish ON-bipolar cells, a second mechanism regulates depolarization (*Wong and Dowling, 2005*). This is mediated by a group of postsynaptic glutamate transporters from the Solute Carrier (SLC1) family (*Niklaus et al., 2024*). These transporters act as dual-purpose proteins, carrying glutamate, three Na +, and one H+ inside the cell and counter-transporting K+ (*Fahlke et al., 2016*), consequently generating a thermodynamically uncoupled current by Cl- flow (*Fairman et al., 1995*; *Mim et al., 2005*), able to control ON-response to light along with the human homolog mGluR6b. Zebrafish *eaats* complement features 11 members distributed at different locations in the central nervous system (*Gesemann et al., 2010*; *Niklaus et al., 2017*). Of these, only EAAT5b and EAAT7 are located in the dendritic tips contacting the majority of photoreceptors (*Niklaus et al., 2024*). Electroretinogram experiments on *eaat5b*$^{-/-}$, *eaat7*$^{-/-}$ and double mutant larvae exposed to wide field white light show that the two transporters impact differentially ON-BCs depolarization in both amplitude and dynamics. Namely, lack of EAAT5b causes a modest lowering of the max amplitude and a steep reduction of the time-to-peak of the wave, while loss of *eaat7* does not lead to an amplitude effect and makes the ON-BCs depolarize slower (*Niklaus et al., 2024*). Previous studies indicate that mGluR6b-dependent ON-response might have a wavelength-dependent tuning (*Saszik et al., 2002*; *Nelson and Singla, 2009*). By performing monochromatic ERG (*Zang et al., 2015*), we observed that lack of either EAAT5b or EAAT7 leads to moderate defects in the ON-response only under certain wavelength ranges. Interestingly, lack of either Eaat5b or Eaat7 leads to long-wavelength response defects only. Conversely, when both factors are absent, the defect becomes observable for short-wavelengths responses as well. Analysis of the distribution of these transporters through the whole larval outer plexiform layer (OPL) revealed a complementary pattern for the two postsynaptic Eaats, where EAAT5b appears to concentrate in the 'area temporalis' or 'strike zone' (SZ), an important region for prey detection and capture (*Zimmermann et al., 2018*; *Yoshimatsu et al., 2020*). We developed a behavioral assay based on UV-light stimulation to evaluate the hunting response in our mutants and found no significant differences in *eaat5b* mutants. However, *eaat7* mutants displayed

a significant increase in responses to our stimulation, suggesting a role of these genes in regulating UV-signal integration for prey capture. Optomotor response (OMR) analysis targeting red cones demonstrated a defect in motion detection in *eaat5b-/-* larvae, whereas *eaat7-/-* mutants exhibited no detectable defects. These findings suggest that EAAT5b and EAAT7 regulate distinct aspects of ON signal transmission dynamics at the level of bipolar cells (BCs), with mGluR6b acting as the primary regulator of response amplitude.

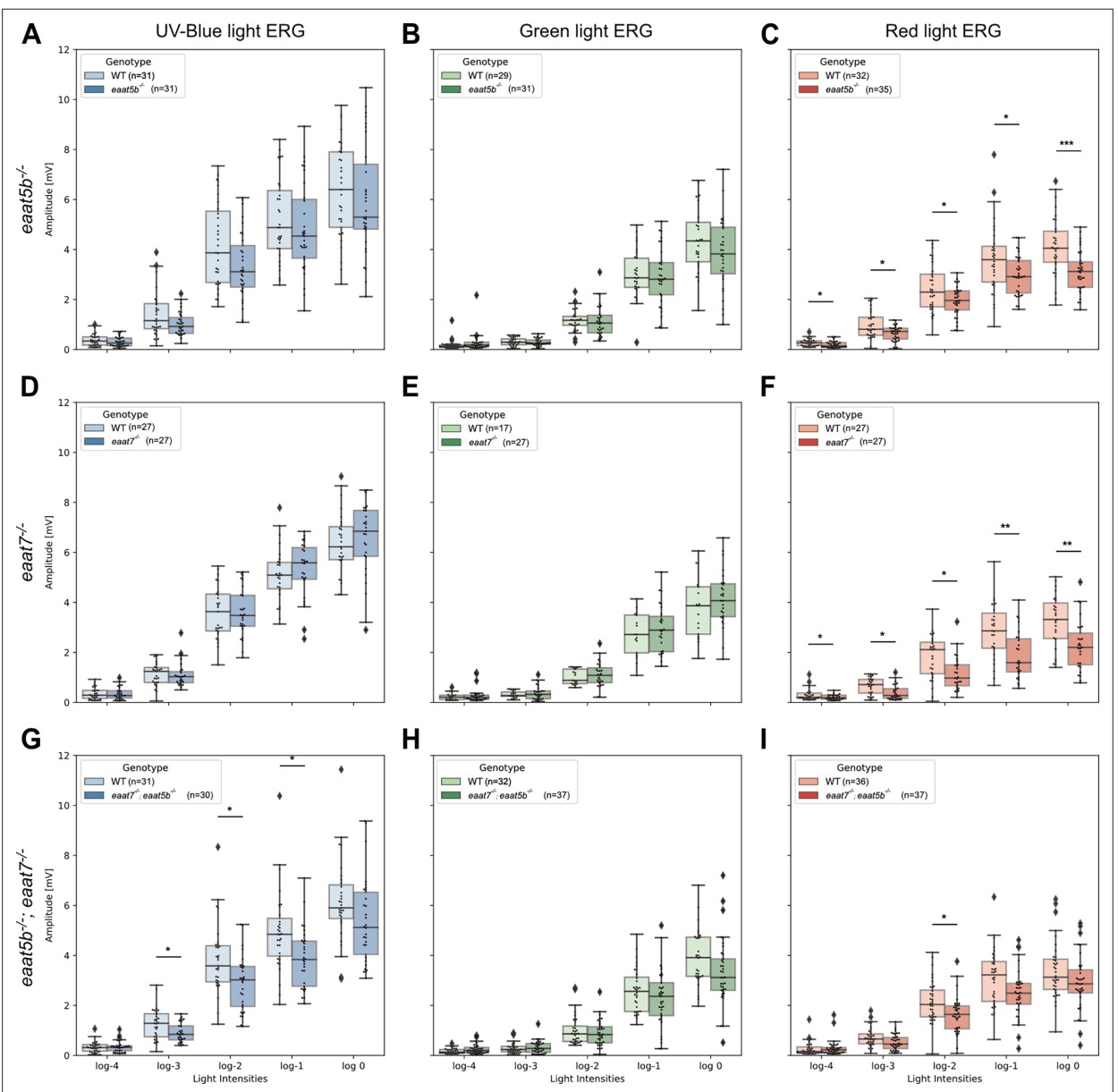

**Figure 1.** EAAT5b and EAAT7 mediate UV-blue and red but not green light signal transmission. Monochromatic electroretinograms (ERG) was recorded on *eaat5b-/-* (**A-C**), *eaat7-/-* (**D-F**) and *eaat5b-/-; eaat7-/-* (**G-I**). B-wave amplitudes are plotted in box-and-whisker plots with fish-specific data overlaid as a swarmplot. Homozygous mutation in *eaat5b* or in *eaat7* causes a decrease in the b-wave if stimulated by red light (**C** and **F**) but not if stimulated with green or UV-blue light (**A, B, D, and E**). Double KO animals display a reduced b-wave amplitude for UV-blue (**G**) and red light (**I**) but not for green light (**H**).

## Results

### EAAT5b and EAAT7 present spectral-specific inputs to the ON-response

Previous publications studying the spectral characteristics of ON-response regulation suggested that both metabotropic and non-metabotropic components of this machinery might possess wavelength-specific responses (*Saszik et al., 2002*; *Nelson and Singla, 2009*). To assess whether EAAT5b and EAAT7 show any chromatic tuning, we tested mutants in these two *eaat* genes with color-channel-specific ERG recordings (*Zang et al., 2015*; *Niklaus et al., 2024*). Interestingly, neither UV-blue nor green light stimulation led to any significant effect on the b-wave amplitude (proxy of ON-BCs depolarization) between both *eaat5b* and *eaat7* KO animals when compared to WT (*Figure 1A–B; D-E*). Red light stimulation, however, caused a decreased b-wave amplitude throughout all five log units in both *eaat5b*[-/-] (adjusted p-values of 0.0331, 0.0127, 0.0331, 0.0331, 0.0009, respectively) and *eaat7*[-/-] (adjusted p-values of 0.037, 0.013, 0.011, 0.005, 0.006, respectively) larvae (*Figure 1C–F*). Monochromatic ERG recordings on double *eaat5b*[-/-]*; eaat7*[-/-] mutant fish displayed a decreased b-wave amplitude both when stimulated with UV-blue (log-3: 0.024 adjusted p-value, log-2: 0.022 adjusted p-value, log-1: 0.022 adjusted p-value) as well as red (log-2: 0.023 adjusted p-value) light (*Figure 1G–I*). This suggests that neither EAAT5b nor EAAT7 participate in green light signal transmission, but one of these transporters might be involved in transmitting short-wavelength light information, originating from either blue or UV cones.

### Distribution of Eaat5b and Eaat7 partially reflects cones anisotropies

We set out to examine the expression patterns and locations of these two proteins across the outer plexiform layer in order to explore the mechanisms behind these wavelength-specific responses. In a previous study, we have shown that EAAT5b and EAAT7 co-localize at the dendritic terminal of bipolar cells in contact with all the cone types and with some rods in the adult retina (*Niklaus et al., 2024*). While the zebrafish adult photoreceptor cell layer displays a rather homogeneous mosaic structure (*Allison et al., 2010*), the larval eye features strong anisotropies in the distribution of the different cone types (*Zimmermann et al., 2018*). In particular, UV cones are strongly enriched in the 'strike zone' (SZ), a temporo-ventral region of the retina where this photoreceptor subtype displays morphological and functional changes resulting in higher sensitivity and slower signal integration dynamics (*Zimmermann et al., 2018*; *Yoshimatsu et al., 2020*). We performed whole mount immunohistochemistry on 6 dpf dissected retinas, imaging most of the OPL and recording EAAT5b, EAAT7 and both UV and red cone pedicles in order to document any EAAT5b and EAAT7 and to test if these transporters' distribution is biased towards one or more cone types. Maximum projections of WT retinas show that these two proteins appear to co-localize along the whole retina (*Figure 2A*) as previously observed (*Niklaus et al., 2024*). However, the relative intensity of each transporter changes in different regions of the OPL, with EAAT7 intensity dropping in the temporal area of the eye, where instead EAAT5b signal increases (*Figure 2A* dashed line). To further investigate this uneven distribution, we selected virtual midsections (see Methods) from the recorded eyes and calculated the relative intensities of EAAT5b, EAAT7 and UV/red cone pedicles, respectively, starting each measurement from the ventral part of the SZ (*Figure 2B*). Virtual midsections show that EAAT5b signal reaches peak intensity in the SZ, while EAAT7 shows the highest signal in the nasal patch of the retina. As observed by the UV cones staining and previously published work (*Yoshimatsu et al., 2020*), UV cone pedicles are concentrated exactly in the peak intensity regions of both EAAT5b and EAAT7 (*Figure 2C*). Relative intensity plots, pooling the whole set of recorded retinas (n=11), confirm what was already observed from the microscope images, further highlighting the similarities in the EAAT5b and UV cones recorded traces and pointing out the markedly higher intensity of EAAT7 in the nasal region of the retina (*Figure 2D* top panel). A correlation matrix of all the traces shows that UV cones strongly correlate with EAAT5b, while EAAT7 shows lower similarities (*Figure 2D* bottom panel). We successively investigated the retinas from the red cones reporter line (*Figure 2E*). These photoreceptors mostly populate the dorsal half of the retina, becoming scarcer close to the SZ (*Figure 2E*). EAAT5b and EAAT7 traces showed similar intensity patterns to what was observed in the previous experiment, but neither of the two proteins distribution fitted the intensity profile of the cone pedicles as previously seen for UV cones and EAAT5b (*Figure 2F* Top). However, the correlation matrix indicates high similarity between EAAT7 and red cone pedicles (*Figure 2F* bottom panel). The discovery of the existing anisotropy between

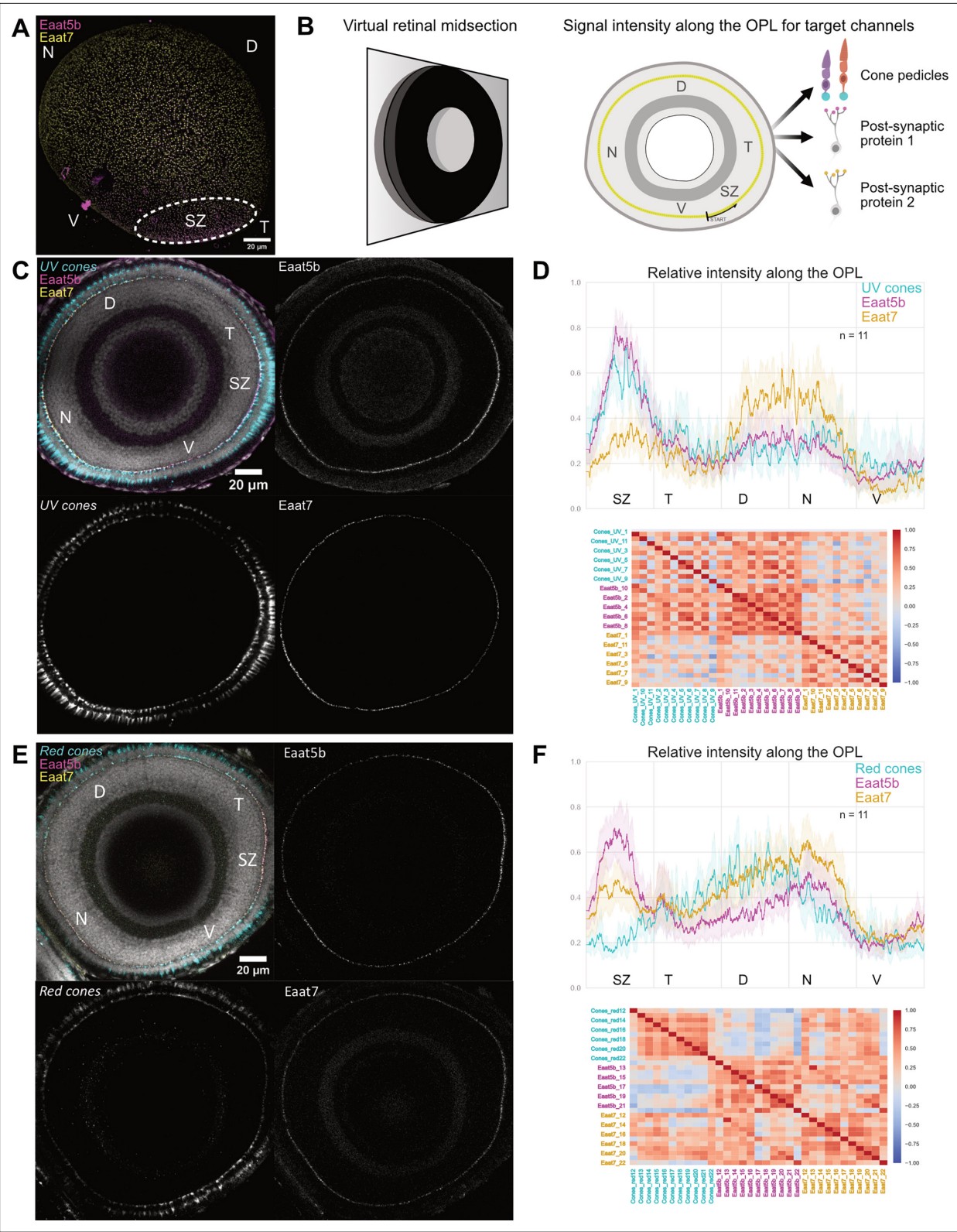

**Figure 2.** EAAT5b distributes anisotropically to EAAT7 and focuses in the Strike Zone in contact with UV cones. (**A**) Projection of the 6 dpf whole larval retina stained for EAAT5b (magenta) and EAAT7 (yellow) shows that both proteins appear co-expressed along most of the outer plexiform layer (OPL) apart from the temporo-ventral area called strike zone (SZ) where relative intensity of EAAT7 drops while EAAT5b raises, indicating distribution anisotropy. (**B**) Schematics depicting the process used to measure the relative intensity of fluorescent signals along the OPL. Left: One section (retinal

*Figure 2 continued on next page*

*Figure 2 continued*

midsection) where most of the photoreceptor layer is perpendicular to the z-axis is selected. Right: on the selected section, the signal intensity for the different channel is measured by tracing a line along the visible OPL starting from the ventral extreme of the SZ and the average gray level under the trace is recorded and normalized to have values between zero and one for all channels. (**C**) Virtual midsection of a 6 dpf WT retina with nuclei (gray), UV cones (cyan), EAAT5b (magenta), and EAAT7 (yellow) labeled. (**D**) Top: Relative intensity values of the three fluorescent channels along the whole OPL. Full-colored line represents the traces' mean and semitransparent area represents 95% CI. Bottom: Pearson's correlation matrix pooling all the measured traces. While the overall correlation score is relatively high, probably due to partial co-localization, EAAT5b and UV cone pedicles appear to define a cluster on their own when compared to Eaat7 traces. (**E**) Virtual midsection of a 6 dpf WT retina with nuclei (gray), Red cones (cyan), EAAT5b (magenta), and EAAT7 (yellow) labeled. (**F**) Top: Relative intensity values of the three fluorescent channels along the whole OPL. Full-colored line represents the traces' mean and semitransparent area represents 95% CI. Bottom: Pearson's correlation matrix pooling all the measured traces. Notably, while EAAT7 and EAAT5b still correlate with a high score amongst each other, the former's traces appear to be more similar to red cone pedicles compared to EAAT5b. This is probably due to the differences at the Strike Zone. For the whole panel, 'N:' nasal, 'D:' dorsal, 'T:' temporal, 'V:' ventral, 'SZ:' strike zone.

The online version of this article includes the following figure supplement(s) for figure 2:

**Figure supplement 1.** mGluR6b outer plexiform layer (OPL) distribution peaks in the strike zone (SZ) in a similar pattern to Eaat5b.

EAAT5b and EAAT7 led us to inquire into mGluR6b distribution in the OPL. Preliminary data (N=1) shows that the metabotropic receptor's distribution along the retina follows a similar, if even narrower, pattern as the one shown by EAAT5b (*Figure 2—figure supplement 1*). We know from previous publications that the use of an mGluR6b agonist drug can lead to monochromatic ERG defects that are stronger for short wavelength stimulation compared to the rest of the spectrum (*Saszik et al., 2002*). It is possible that this defect would stem from a higher concentration of mGluR6b in the dendritic terminals contacting UV cones, consequently more affected by the block. Taken together, these results suggest that EAAT5b and EAAT7 have different functions in the anisotropic larval retina. Even if both localize in contact with UV and red cones, our data suggest a difference in expression ratio between the two, indicating subspecialized roles in the different retinal compartments. UV cones in the SZ are crucially important for successful prey detection. Previous studies hypothesize that UV cones residing in the nasal patch are relevant for predator avoidance (*Yoshimatsu et al., 2020*). This suggests that the different biophysical properties previously observed in EAAT5b and EAAT7 (*Niklaus et al., 2024*) might aid the information acquisition for one or the other behavior.

## Zebrafish display hunting response in an intensity-dependent trend

Both zebrafish short wavelengths cones (UV and blue) have mostly overlapping wavelength-detection ranges (*Yoshimatsu et al., 2021*). This makes isolation between these two cone populations using a wide field stimulus non-trivial. Larval hunting strongly relies on the specialized UV cones residing in the SZ and their ablation drastically impairs prey detection as seen in published behavioral assays (*Yoshimatsu et al., 2020*). Our previous results point towards EAAT5b having a role in UV signals integration and its peculiar concentration in the SZ might be explained by a function in prey detection signaling. Testing differences in sensitivity with this innate larval behavior could help differentiate between a functional defect stemming from either UV or blue cones. Therefore, we built a behavioral setup to evoke a hunting response in zebrafish larvae using an artificial and programmable UV stimulus. The setup records the behavior of a single larva head-mounted in an arena with a screen where a single UV dot is projected from a focused LED source (*Figure 3A*). The UV LED is inserted in a scaffold holding lenses and a pinhole that focuses the light in a single small dot roughly the size of a paramecium (~4.9°) (*Figure 3B*). A servomotor at the base of the LED scaffold provides movement to the stimulus and both movement and LED control are defined by a computer-triggered Arduino (*Figure 3C*). Eye convergence and tail flick events can be then recorded and identified as hunting events (*Figure 3D*). Virtual hunting assays have been previously used to have a higher control over stimulation through the use of projectors displaying white dots on a darker background (*Bianco et al., 2011*; *Semmelhack et al., 2014*). In particular, a recent work showed that UV light prominently evokes prey-capture behavior in zebrafish larvae (*Khan et al., 2023*). Due to the nuances of our setup, we decided to validate it by exposing WT larvae to max intensity stimuli with LEDs that would target shorter and longer wavelength cones, respectively (*Figure 3E*). 6 dpf WT larvae were exposed to ten rounds of stimulation at max intensity, consisting of a leftward and rightward stimulus in succession. The same larva was exposed first to orange light and successively to UV, following the same stimulation pattern. As expected, zebrafish larvae strongly react to a virtual hunting stimulus in the UV wavelength range

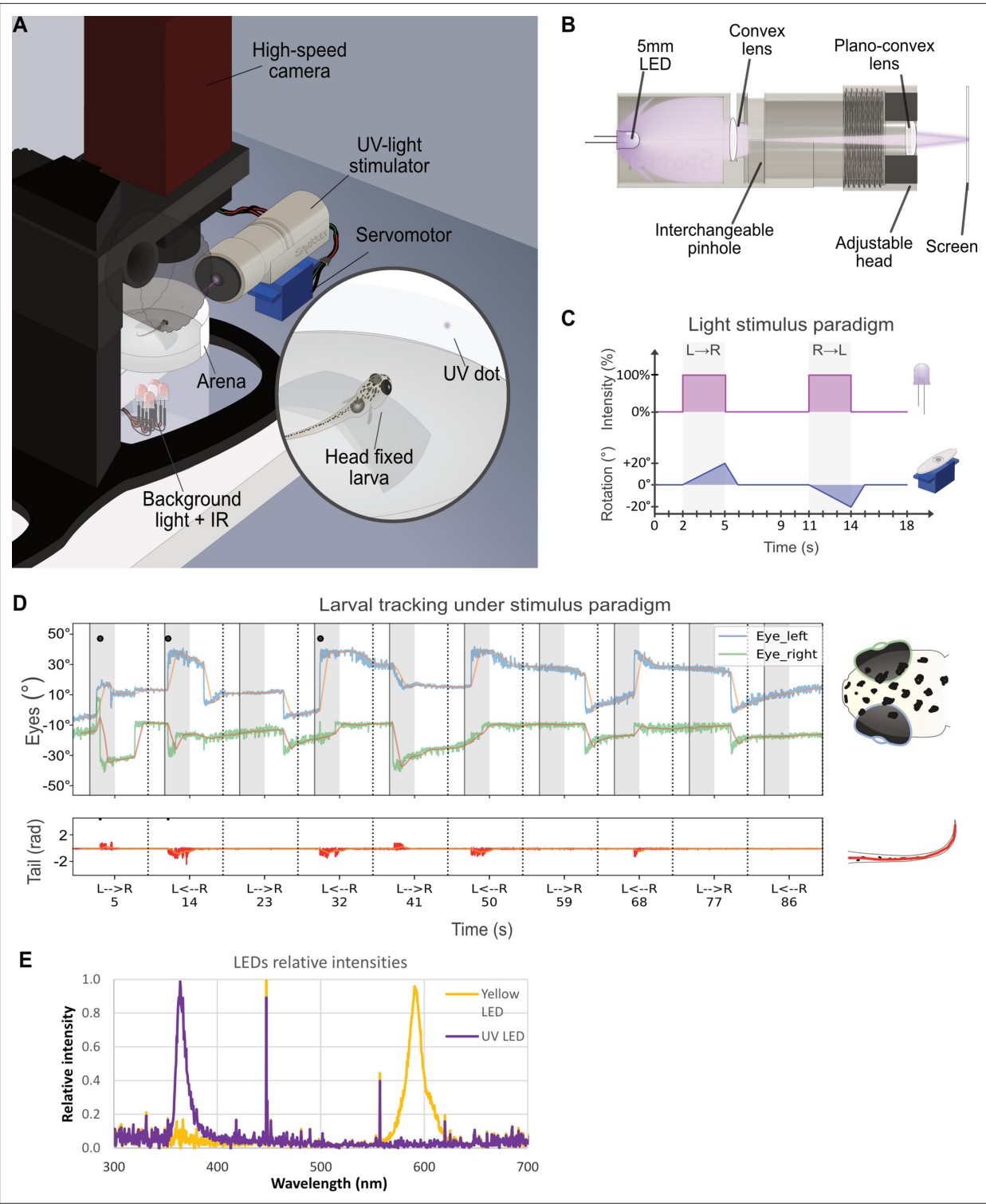

**Figure 3.** UV stimuli can be used to simulate UV-bright moving paramecia in a LED-based custom setup. (**A**) Schematics of the custom setup used to deliver virtual prey-capture stimuli. (**B**) Schematics representing a section of the UV-light stimulator and approximate light pathway through pinholes and lenses. (**C**) Diagram representing the stimulus paradigm during one iteration. After 2 s from the start of the round, the LED turns ON along with the servomotor, which turns left to right by 20° (~90° on the projected screen). This takes 3 s, after which the LED turns OFF and the servomotor resets position. The same operation reiterates in the opposite direction after 4 s. (**D**) Sample tracking graphs from the setup. Top panel represents eye-specific angle variations over time (blue and green) with rolling averages used to identify events in orange. Positive increases denote left eye convergence, while negative increases denote right eye convergence. The bottom panel represents tail total curvature over time (red) with rolling average used to identify

*Figure 3 continued on next page*

*Figure 3 continued*

events in orange. Black dots represent a detected hunting event. (**E**) Relative intensities of the UV and yellow LEDs used for the validation measured by spectrometer.

(*Figure 4B*), but have very limited response to light targeting the middle-/long-wavelength cones (*Figure 4A*). Interestingly, larvae showed an initial higher response, waning off by the 10th repetition, probably due to stimulus habituation. We next tested if the UV stimulus detection was intensity-dependent and if a 'detection threshold' could be identified. For this, we exposed WT 6 dpf larvae to an increasing intensity paradigm, where the 10 rounds would expose the zebrafish to a range of intensities starting from 0 to 90% (*Figure 4B*). Following our expectation, the number of recorded hunting responses steadily increased along with the stimulus strength from intensity 40%, reaching a plateau around intensity 70%.

In sum, we successfully built a behavioral setup that can stimulate zebrafish larvae with a program-mable UV prey-like stimulus. We validated the machine following a similar paradigm already done with live prey (*Yoshimatsu et al., 2020*) and observed that larvae strongly react to UV light compared to orange. Moreover, we could demonstrate that number of responses is dependent on the stimulus intensity and that with our setup, WT larvae can generally detect the stimulus starting from 40% of intensity. This suggests that this setup can be used to test UV sensitivity in our mutant larvae.

## Loss of EAAT7 positively affects prey detection sensitivity

Whole mount retina imaging shows that both EAAT5b and EAAT7 are expressed at the dendritic tips of UV cones, being concentrated in the SZ and in the nasal patch, respectively. Monochromatic ERG recordings show that KO of both *eaat5b* and *eaat7* leads to a reduced b-wave in larvae stimulated with short wavelength light. Both protein localization (the SZ is an essential region for prey capture *Zimmermann et al., 2018*; *Yoshimatsu et al., 2020*) and functional data point towards EAAT5b being involved in UV vision and prey detection. To prove this, we tested *eaat5b*[-/-] mutant larvae with the UV hunting sensitivity assay we previously validated. 6 dpf WT fish (n=25) and *eaat5b*[-/-] mutants (n=23) were exposed to our UV incremental intensity hunting assay (see methods). The animals were tested in alternate order to avoid potential circadian differences and the results were obtained from four different experimental days. Our experiments show that even if scarcely (4%), WT larvae can react to the UV stimulus already with 30% of LED intensity (*Figure 5A*), while mutants lacking EAAT5b start responding from intensity at 40% (*Figure 5B*). While the response peak is reached in both genotypes at 80% intensity, WT larvae have a stimulus detections success of 44% (*Figure 5A*), *eaat5b*[-/-] mutants show a lower response (35%) (*Figure 5B*). Surprisingly, light-intensity-specific comparison between eaat5b[-/-] mutants and WT animals did not indicate any significant difference (Chi-square test *p*-value >0.05 in every instance). Similarly, neither the comparison of the proportion between successful hunting responses over total stimuli through the whole test, nor the one between responses distri-butions across the intensity levels resulted in a significant difference (two proportions z-test p-value: 0.16; Kolmogorov-Smirnov test p-value: 0.83) (*Figure 5C*). This indicates that either our assay is not sensitive enough to detect a diminished UV-prey-detection defect indeed there is no such defect in *eaat5b* mutants. We then proceeded by testing *eaat7* mutants using the same paradigm. Inter-estingly, *eaat7* mutants (n=19) displayed overall better responses when compared to the WT larvae (n=18) (*Figure 6* right panel). While the latter could still detect the stimulus from intensity 30% (~5%) (*Figure 6A* left), *eaat7*[-/-] fish started reacting to UV from intensity 20% (~5%) (*Figure 6B* left panel). Response peak is reached at 60% for the controls and between 80 and 90% for the mutants; with WT larvae with a response of 33% while *eaat7* mutants reach 53% of response (*Figure 6* left panel). As with eaat5b[-/-] mutants, light-intensity-specific comparison between eaat7[-/-] mutants and WT animals did not indicate any significant difference (Chi-square test *P*-value >0.05 in every instance). By comparing pooled responses over the total of delivered stimuli, we discovered that WT animals performed less than *eaat7* mutants did (two proportions z-test p-value: 0.0011; *Figure 6C* left panel). Conversely, the comparison of intensity-specific responses distributions led to no significant difference (Kolmogorov-Smirnov test p-value: 0.17) (*Figure 6C* right panel). Taken together, our assay shows that lack or presence of both the postsynaptic EAATs might impact response to UV light and prey capture and, interestingly, the lack of EAAT7 appears to facilitate prey detection. Previous studies from our group (*Niklaus et al., 2024*) show that EAAT5b and EAAT7 have different gating dynamics along with

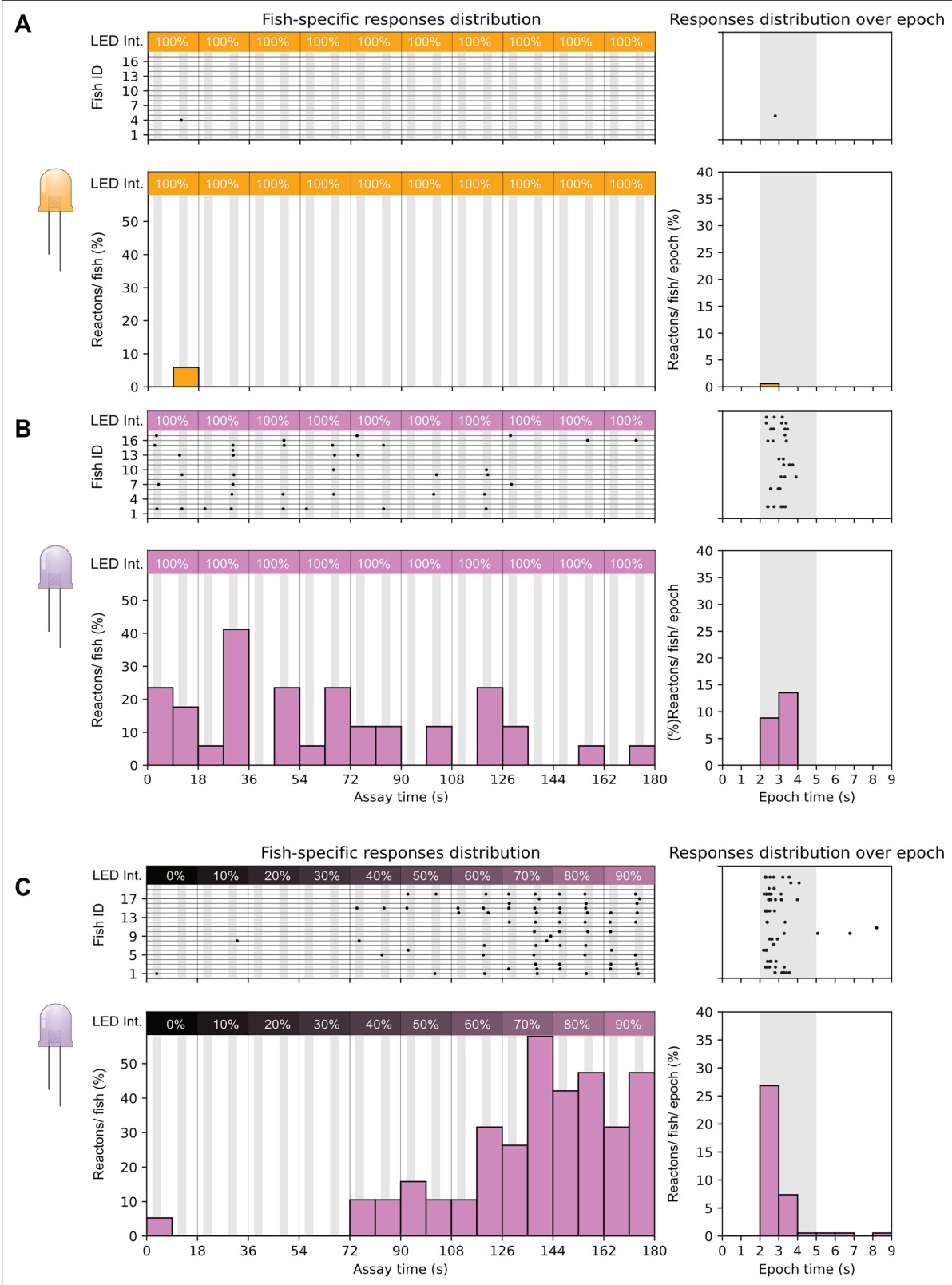

**Figure 4.** Larval zebrafish show hunting response preferentially to UV stimuli in an intensity-dependent trend. 6 dpf WT zebrafish larvae were exposed to yellow (**A**) and UV (**B**) full intensity hunting stimuli. While response was extremely low for the yellow light (1 response overall; **A**), UV stimuli elicited consistent response in the first stimulation rounds, gradually waning through each repetition (**B**). Wild-type (WT) larvae exposed to incremental UV light intensity showed increasing response rates, reaching a response plateau between the last two intensity levels (**C**). For all graphs in the figure: top shows

*Figure 4 continued on next page*

Figure 4 continued

fish-specific results through the whole assay (left) and pooled events distribution over single epoch (right); bottom shows summary statistics as response percentages (reactions/fish on the left, reaction/fish/epoch on the right).

different chloride current generation capacity. Hence, it is conceivable that the lack of one or the other factor changes signal responses and information integration in important areas such as the SZ, thus altering the larva's ability to detect and respond to a prey stimulus.

## *eaat5b*[-/-] but not *eaat7*[-/-] fish have a lower OMR response to red and black bands

Our monochromatic ERGs showed that lack of either EAAT5b or EAAT7 leads to a lower b-wave generation upon red light stimulation. To confirm this result, we designed a high-throughput OMR assay targeting the zebrafish long-wavelength cones by exposing 5 dpf larvae to moving gratings in black and red in a range of different contrasts delivered in pseudo-random order (*Figure 7A–B*). Fish were exposed to two stimuli, one still (habituation phase) and one moving towards the right (stimulus phase), alternated by black and white centering stimuli to reposition the larvae at the center of the arena (*Figure 7B*). Next, we calculated each fish's rightward movement percentage during both habituation and stimulus phases to observe whether EAATs mutants would perform worse than their WT controls. While during the habituation phase movement direction remains neutral (~50% to either side), all fish displayed a rightward movement increasing along with the delivered black-vs-red contrast during the stimulus phase (*Figure 7C*). As expected, *eaat5b* mutants show a marked reduction on the time spent swimming towards the stimulus, particularly at the higher contrasts (contrast: 213; $p$=0.01) (*Figure 7C*). Interestingly, *eaat7*[-/-] fish do not show any visible defect when compared with WT even if monochromatic ERG data showed a marked decrease in their b-wave. Both mutants increase overall movement in a contrast-dependent trend with no significant differences from the respective WT control groups (*Figure 7—figure supplement 1*). The optomotor response draws information from both the ON and OFF pathway, and it is conceivable that removing EAAT7 would not generate a defect large enough to be measured by our assay.

## Discussion

In this study, we aimed to investigate the functional role of EAATs expressed on ON-bipolar cells in the retina. These transporters are responsible for clearing glutamate from the synaptic cleft while simultaneously inducing an anion conductance that hyperpolarizes the expressing cell in the presence of glutamate. As a result, they serve a dual function as both glutamate transporters and inhibitory glutamate receptors (REF). Our previous studies demonstrated that the loss of EAAT5b and EAAT7 reduces ON-bipolar cell responses, with eaat5b mutants being less affected than eaat5b/eaat7 double mutants, which also exhibit altered response kinetics (*Niklaus et al., 2024*).

In order to gain more insight into the modulatory role of these transporters and their effect on vision, we first assessed the electroretinographic responses to a variety of spectral illumination.

We found that a lack of EAAT5b and EAAT7 leads to modest reductions in the ON-response under UV-blue and red light. Interestingly, we did not observe any reduced responses under green illumination, suggesting a spectral-specific role in modulation by these proteins. However, due to the strong overlap of UV and blue cones' light sensitivities, our whole retina recordings with a wide-field stimulus cannot separate these two channels (*Yoshimatsu et al., 2021*).

Recent studies have uncovered a complex anisotropy of the larval zebrafish eye, matched to the visual ecology that they live in *Zimmermann et al., 2018*. This adds a complexity of visual processing that cannot be assessed by the whole-field stimulation of our experiments.

In order to assess anisotropic distribution of EAAT5b and EAAT7 in the retina, we performed immunohistochemistry on larval dissected retinas. In line with previous data, we observed that EAAT5b and EAAT7 are largely co-expressed at most dendritic tips (*Niklaus et al., 2024*). More intriguingly, we found an unequal density of immunoreactive across the retina. EAAT7 is highest expressed in ventral regions of the retina with an intensity drop at the area temporalis or the strike zone (SZ). Conversely, EAAT5b's signal is elevated in this specific area, where the highest concentration of UV cones is also

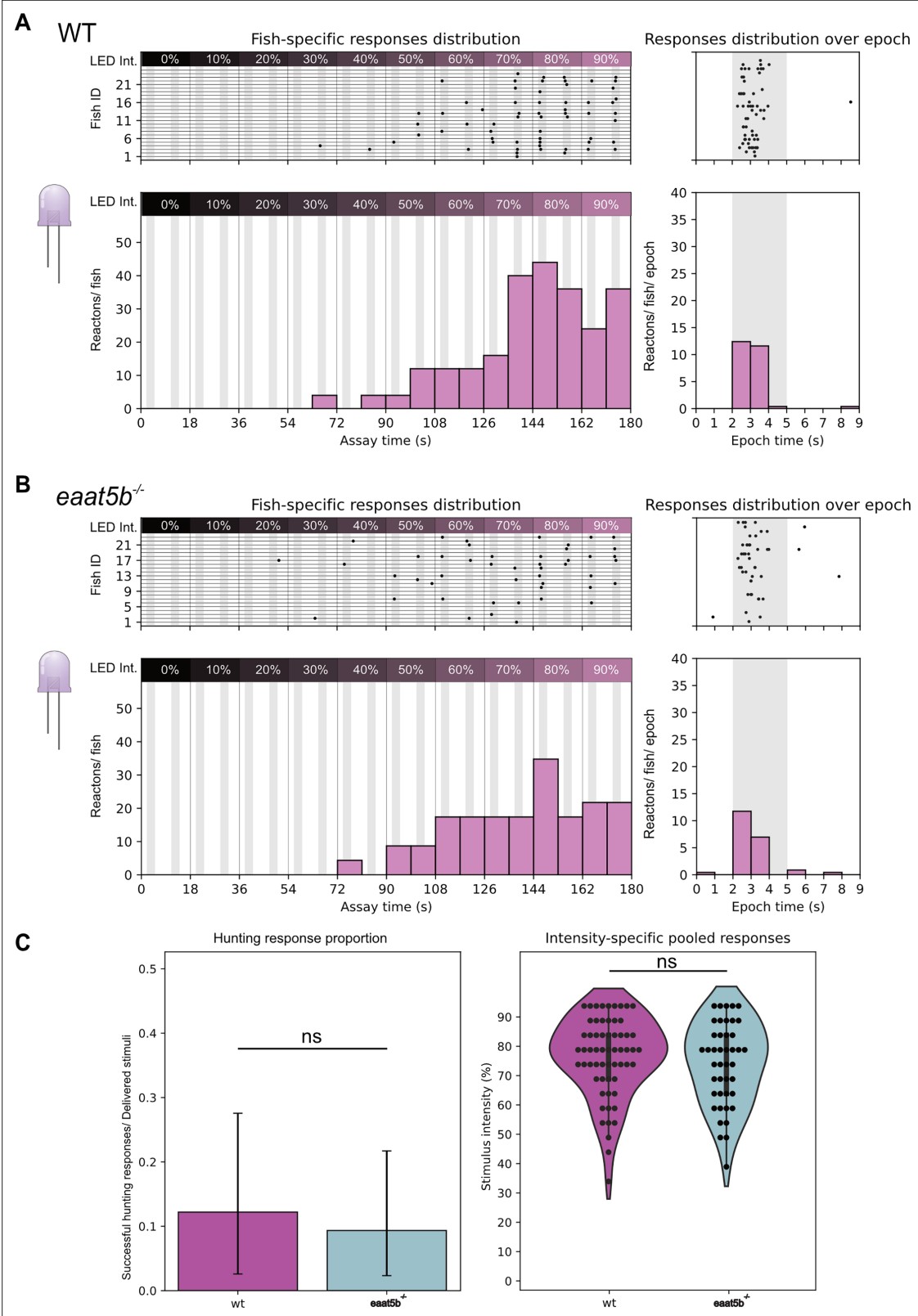

**Figure 5.** *eaat5b* KO mutants do not show a significant difference in UV hunting response compared to wild-type (WT). *eaat5b* mutants (n=23) begin reacting to the incremental prey stimulus one intensity step later than WT (n=25) and display overall lower reactions/fish. While WT larvae start showing responses with stimuli delivered already at 30% of intensity (**A**), *eaat5b* mutants start showing reactions from 40% (**B** left). WT larvae reach peak responses at intensity 80% with a markedly higher response (**A**) compared to *eaat5b* mutants (**B**). Pooled responses proportion does not show any

*Figure 5 continued on next page*

*Figure 5 continued*

significant difference between WT and *eaat5b* mutant fish (two proportions z-test p>0.05) (**C** left). Comparing stimulus intensity-specific responses also leads to a non-significant difference (Kolmogorov-Smirnov test p>0.05) (**C** right). For all graphs in the figure: top shows fish-specific results through the whole assay (left) and pooled events distribution over single epoch (right); bottom shows summary statistics as total response proportions (successful hunting responses/ delivered stimuli) and intensity-specific percentages (reaction percentage/ stimulus intensity).

found. This zone is aptly named for its importance in prey detection (*Zimmermann et al., 2018*; *Yoshimatsu et al., 2020*).

Previous work investigating the importance of UV cones for prey-capture circuits proposed that while UV-sensitive photoreceptors in the SZ are important for hunting, morphologically similar cones in the nasal patch might be used to monitor the periphery for predators (*Yoshimatsu et al., 2020*).

The peculiar localization of Eaat5b in the SZ, along with the observed defect in the UV/Blue range upon KO, suggests an involvement of Eaat5b in prey detection and capture. Hunting behavior has been extensively studied in the past years, both using live paramecia (*Semmelhack et al., 2014*; *Yoshimatsu et al., 2020*) and virtual assays (*Bianco et al., 2011*; *Semmelhack et al., 2014*), where generally a moving white dot is projected on a screen and displayed to a larva. In order to test UV-specific sensitivity in our mutants, we built a behavioral setup to deliver UV-bright artificial stimuli simulating swimming paramecia to head-mounted larvae. The setup delivers programmable stimuli at different light intensities and can be used to determine wavelength detection thresholds. We validated the setup by testing WT fish reactions to yellow and UV light, first confirming that responses to UV light were significantly more frequent than those to yellow light, and second, that the UV response rate was directly proportional to light intensity. Surprisingly, we did not observe a significant effect on prey-detection and capture behavior in the Eaat5b mutants, while Eaat7 mutants showed an increased response rate and a lower threshold than WT fish. These unexpected results may be accounted for by the different dynamics of Eaat5b and Eaat7. Namely, removal of Eaat5b leads to a shortening of the time to peak of the measured b-waves, while Eaat7 causes a lengthening of the same, indicating opposing roles in regulating the ON-BCs depolarization timing (*Niklaus et al., 2024*). UV-cones in the SZ have longer signal integration compared to the other photoreceptors, supposedly to better track a moving prey (*Yoshimatsu et al., 2020*). The concentration of Eaat5b in the BC terminals of this region might be another mechanism to slow down the signal and facilitate this behavior. Loss of Eaat5b might then result in a reduced prey-tracking ability; while conversely, the absence of a functional Eaat7 could lead to slower integration times throughout the whole retina, consequently making stimulus sensitivity higher. A possible way to test his hypothesis would be to modify our existing set-up to display the moving dot approaching from the periphery of the larval eye and observing if the animal's 'detection field' in absence of Eaat7 has expanded outside of the SZ.

Since our monochromatic ERG revealed long-wavelengths sensitivity defects in both Eaat5b and Eaat7 mutants, we designed a behavioral setup to expose our mutant larvae to OMR-eliciting moving gratings, featuring black and red bands at varying contrasts. Our tests revealed that while Eaat5b mutants exhibited a reduced response to the stimulus, Eaat7 mutants showed no significant difference compared to WT. Since OMR is driven by both ON- and OFF-stimuli (*Orger and Baier, 2005*; *Kist and Portugues, 2019*) and zebrafish larvae typically respond equally to both contrast changes when their field of view is restricted (as in our assay), it is possible that any impairment in the ON-response was compensated by the larvae's strong reaction to light-dark transitions.

Behavioral assays can be easily structured to deliver high-throughput results with intuitively measurable effects. However, these results are the outcome of the whole neural circuit, making it difficult to assess complex mechanisms without more targeted assays. Both Eaat5b and Eaat7 transcripts are expressed throughout the brain (*Niklaus et al., 2024*), and both localization and effects of their loss in the zebrafish brain are yet to be investigated. Combining monochromatic ERGs with full retinal immunohistochemistry and behavioral experiments finally helped us understand more about the functions and differences of these two postsynaptic transporters. Taken together, our data suggests that while mGluR6b remains the main regulator of ON-BCs depolarization throughout the retina, Eaat5b and Eaat7 modulate response speeds in different regions of the retina to optimize behavior-specific integration dynamics, particularly in relation to UV light. In the future, analysis of published single-cell transcriptomics databases could help identify whether *eaat5b* and *eaat7* are co-localized in all ON-bipolar cells or if they are, at least in some cases, segregated in different cell populations.

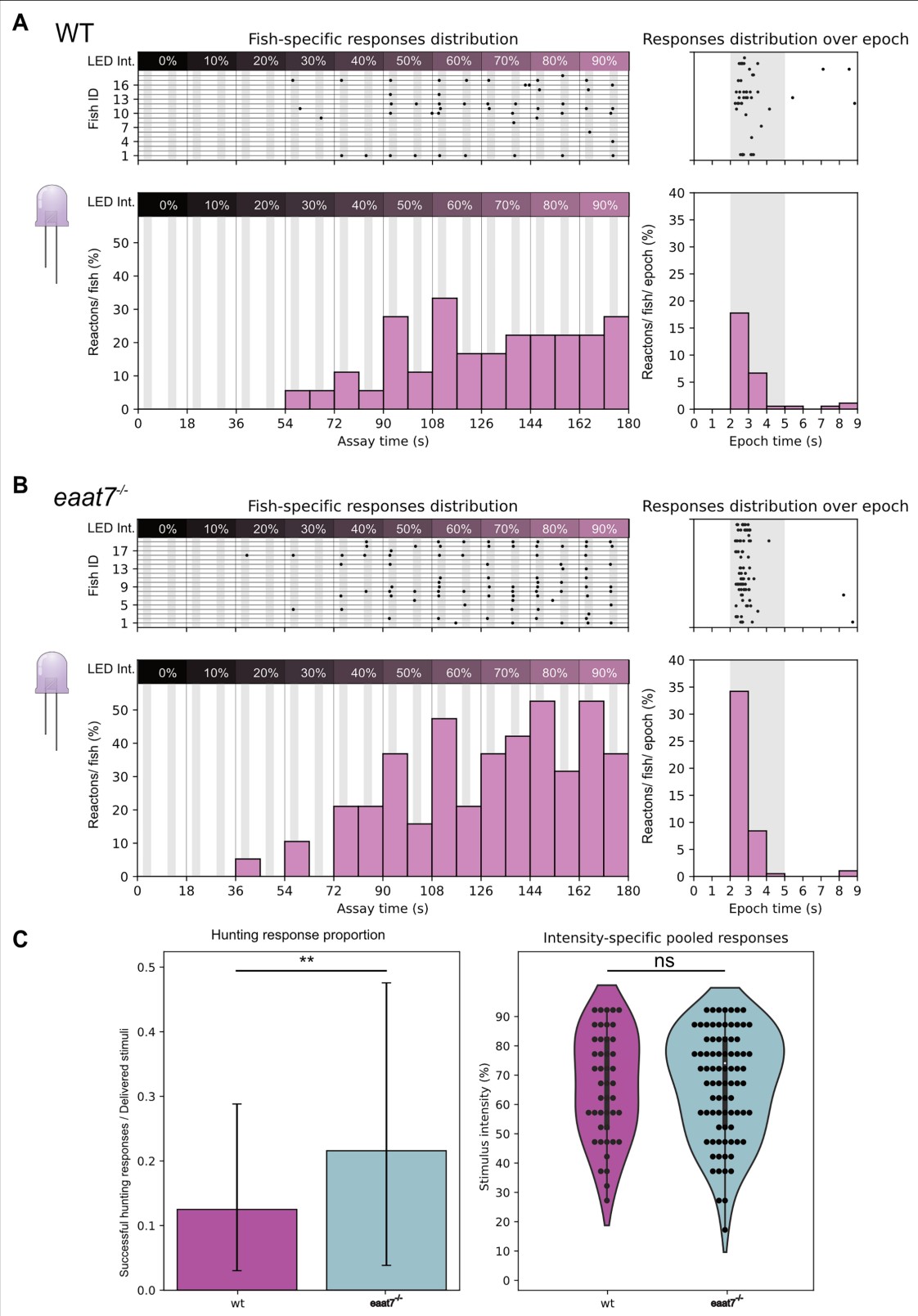

**Figure 6.** *eaat7* KO mutants exhibit a higher peak response to an artificial UV prey stimulus compared to wild-type (WT). *eaat7* mutants begin reacting to the incremental prey stimulus one intensity step later than WT and display overall higher reactions/fish. While WT larvae start showing responses with stimuli delivered at 30% of intensity (**A** left), *eaat7* mutants start showing reactions from 20% (**B** left). Both WT and *eaat7* mutant larvae reach peak responses at intensity 90% but surprisingly, fish lacking the glutamate transporter show a markedly higher response (**A** left) compared to WT (**B** left).

*Figure 6 continued on next page*

*Figure 6 continued*

Pooled responses proportion is significantly increased in *eaat7* mutant fish compared to WT (two proportions z-test p=0.0011) (**C** left). Comparing stimulus intensity-specific responses leads to a non-significant difference instead (Kolmogorov-Smirnov test p>0.05) (**C** right). For all graphs in the figure: top shows fish-specific results through the whole assay (left) and pooled events distribution over single epoch (right); bottom shows summary statistics as total response proportions (successful hunting responses/ delivered stimuli) and intensity-specific percentages (reaction percentage/ stimulus intensity).

## Materials and methods

### Fish husbandry

Zebrafish (*Danio rerio*) were maintained at a standard 14 hr light:10 hr dark cycle (LD) with light on at 8 am and light off at 10 pm as previously described (*Mullins et al., 1994*). Water temperatures were kept between 26 and 28°C. Fish from the AB and TU wild-type strains were used in our study. Embryos were raised in E3 medium (5 mM NaCl, 0.17 mM KCl, 0.33 mM CaCl2, and 0.33 mM MgSO4) containing 0.01% methylene blue to suppress fungal growth or in 1-phenyl-2-thiourea (PTU, Sigma-Aldrich) if prevention of pigment formation was required. Embryos were collected directly after laying. Embryos were then transferred to the incubation room with normal light cycle (14:10). Both 5 and 6 days post fertilization (dpf) animals were sacrificed in ice water, following the second killing method via freezing at –20°C. The following transgenic lines were used for immunohistochemistry: *Tg(zfSWS1–5.5 A:EGFP)* and *Tg(–0.6opn1lw1-opn1lw2:GFP)*(*Tsujimura et al., 2010*) expressing EGFP in UV and red cones, respectively (*Allison et al., 2010*). The following mutant lines were used for ERGs and behavioral assays: *eaat5b⁻/⁻*, *eaat7⁻/⁻*, and double mutants *eaat5b⁻/⁻; eaat7⁻/⁻* (*Niklaus et al., 2024*).

### Monochromatic electroretinograms

Monochromatic electroretinograms (ERG) of WT and mutant animals were recorded at 5 dpf in a double-blinded manner. Larvae were dark-adapted for at least 30 min and preparations prior to recordings were performed under a dim red light to prevent bleaching of photo pigment. The larval eye was removed and placed on filter paper over an agarose gel. The reference electrode was inserted into the agarose gel and the recording electrode, a glass capillary GC100-10 (Harvard Apparatus, Holliston, USA) with a tip diameter of 20-30 μm filled with E3 was placed on top of the cornea.

A series of monochromatic light stimuli of five different light intensities (log –4 to log 0) were projected to the eyes. Each stimulus was 100 ms with an inter-stimulus interval of 7 s. A high-power xenon light source HPX-2000 (Ocean Optics) was used for monochromatic light stimulation and an additional light source (Philips projection lamp type 6958, 20 V, 250 W in a Liesegang Diafant 250 housing) was used for background adaptations. Different light filters were placed in front of both, stimulation and background light source (see *Table 1*).

### Analysis of ERG data

ERG traces were analyzed using Microsoft Excel and Igor-Pro Software (Wave Metrics). B-wave amplitudes were analyzed as a proxy for ON-bipolar cell depolarization. The first 50 ms of each recording were averaged and taken as baseline values. B-wave amplitudes were statistically compared between WT and mutant animals by multiple two-tailed t-tests with multiple comparisons correction using the Holm-Šídák method. All ERG results are plotted in box-and-whisker plots with box showing the quartiles of the respective datasets while the whiskers extend to show the rest of the distribution, except for points that are determined to be 'outliers' (outside of interquartile range * 1.5) as of default with the 'seaborn' Python library boxplots. The box plots are overlaid with a swarm plot representing single fish amplitudes.

### Immunohistochemistry

6 dpf larvae were raised in 0.0003 g/ml phenylthiourea (PTU), sacrificed by incubation in ice water (0°C) and fixed with 1% trichloroacetic acid (TCA) and 2% Sucrose (in PBS) for 60 min at room temperature (RT). After washing in PBS, eyes were dissected and permeabilized by rinsing with proteinase K solution (0.02 mg*ml in PBS).

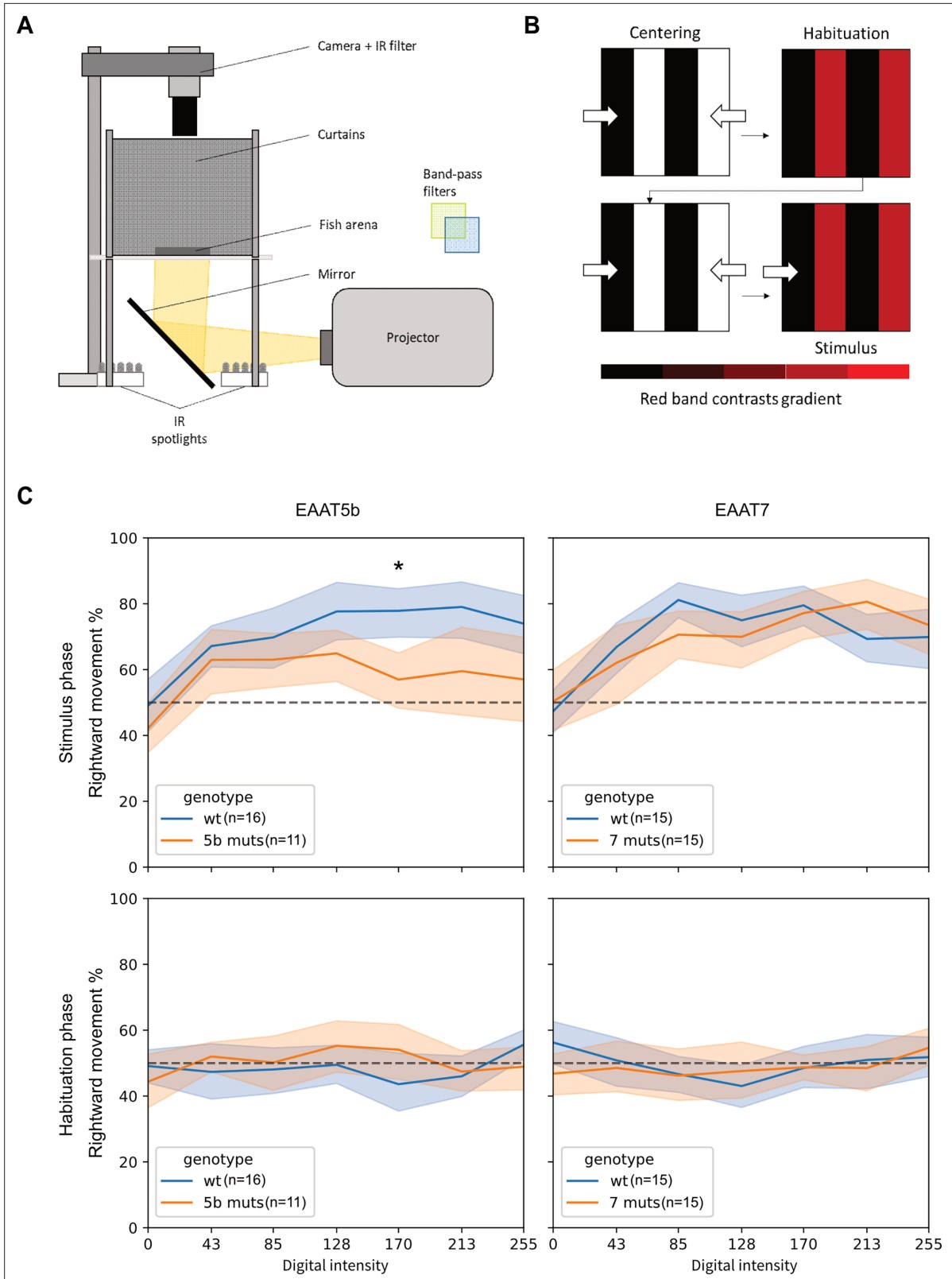

**Figure 7.** *eaat5b* but not *eaat7* mutants show lower response to high contrast long wavelength Optomotor response (OMR). (**A**) To test long-wavelength range sensitivity, we build an OMR setup where a projector sends moving grating at the bottom of an arena organized in rows. Fish swimming in the rows are tracked by a high-speed camera. (**B**) We provide a long wavelength targeting paradigm divided into four steps, where two centering stimuli composed by black and white gratings moving towards the center of the arena, guide the larvae in a central position. After each

*Figure 7 continued on next page*

*Figure 7 continued*

centering stimulus, still (during the Habituation phase) and rightward-moving (during the Stimulus phase) gratings composed of black and red (at different contrasts) bands are displayed and the larval rightward movement percentage over total movement is calculated for both the long-wavelength phases. (**C**) The setup was used for testing sensitivity loss in both *eaat* mutants. As expected, *eaat5b* mutants have an overall reduced response to the stimulus (top left), particularly significant at higher contrasts (Repeated measures ANOVA; $p<0.05$). Conversely, *eaat7* mutants do not show such a defect (top right). In all the tests, fish exposed to a still stimulus (Habituation phase) have no preference independently of the contrast displayed.

The online version of this article includes the following figure supplement(s) for figure 7:

**Figure supplement 1.** *eaat5b* and *eaat7* mutants show the same total activity as their respective control groups during black and red Optomotor response (OMR) assay.

Dissected eyes were post-fixed in 1% TCA, 2% Sucrose in PBS solution for 20 min, washed in PBT (Triton-X100 0.8% in PBS) and blocked in blocking solution (10% normal goat serum, 1% bovine serum albumin in PBT) for 1 hr. Primary antibodies diluted in blocking solution were applied overnight (ON) at 4°C. The following antibodies were used: guinea pig anti-EAAT5b (1:100), rabbit anti-EAAT7 (1:400), chicken anti-GFP (A20162; Invitrogen) (1:500). Primary antibodies were detected with the following secondary antibodies (all 1:500 in PBS) for ON at 4°C: goat anti-rabbit, goat anti-guinea pig, goat anti-chicken and goat anti-mouse all conjugated to Alexa Fluor (AF) 488, 568 or 647 (Invitrogen, Molecular Probes). Cells nuclei were stained with 4',6-diamidino-2-phenylindole (DAPI) (1:1000 in PBS). Eyes were embedded in Mowiol-DABCO and confocal laser scanning imaging was performed with a TCS LSI confocal microscope (Leica). Full retina stacks were recorded with a 40x oil-immersion micro-objective scanning the whole OPL surface in 1 μm steps.

## Image processing and data analysis

Full retina stacks were opened in the software Fiji (ImageJ) (*Schindelin et al., 2012*) and a virtual mid-section, defined as the section in which the three nuclear layers lay perpendicular to the image plane, was selected. Using morphological landmarks in the photoreceptor (UV or Red cones) channel, a segment line was traced along the whole OPL starting from the ventral side of the Strike Zone. The ROI thickness lying under the set line was then enlarged to 20px and the signal intensity lying under the whole ROI was recorded for every channel.

OPL intensities were analyzed through a Python script: traces were imported as Data frame, the photoreceptor, Eaat5b, and Eaat7 channel intensities were normalized over DAPI and divided by trace-specific maximum value. This led to obtain relative intensities with values spanning from 0 to 1. Intensity reading lengths were also normalized to fit between 0 and 1, making each eye comparable with the other. Relative intensities were then smoothed using a rolling average with a window of 25 points (over an average of 3000 data points per trace). Single traces were inserted in a correlation matrix and statistically tested with Pearson's correlation to observe EAAT5b and EAAT7 distribution similarities with both UV and Red cones pedicles.

## Hunting assays

All hunting assays were performed with a custom-built setup composed of a 3.5 cm platform hosting two pairs of infra-red (800–920 nm; OSRAM) LEDs and one white LED (420–650 nm; Elegoo) for tracking camera/background lightning, respectively. An overhead high-speed camera (Pike F032B ASG16; Allied Vision) with a 12mm objective was fixed on an adjustable holder and centered over the platform. At 2 cm from the platform, a custom optics adapter holding a double convex lens (focal length = 22.65 mm; Edmund Optics), a 0.5 mm pinhole wheel, a plano-convex lens (focal length = 10 mm; Spindler&Hoyer) and a 5 mm LED (orange: 550–620 nm; Sander/ultraviolet: 340-380 nm; Roithner LaserTechnik) was fixed on top of a servomotor (SG90; Tower Pro). Both the servomotor and

**Table 1.** Monochromatic ERG light stimulation settings.

| Monochromatic stimulation | Stimulation light filter | Background light filter |
| --- | --- | --- |
| UV-Blue light | 350–430 nm (peak at 415 nm) | 500–680 nm |
| Green light | 450–475 nm | >480 nm |
| Red light | >580 nm | None |

LED were connected to a microcontroller (Arduino UNO; Elegoo) for stimulus control of intensity and movement. Camera and microcontroller were both operated by Stytra (*Štih et al., 2019*), a Python-based open-source behavioral software able to track several behaviors and administer a series of programmed stimuli. 6 dpf larvae were mounted on an arena built from a 3.5 cm Petri dish cap. The forward half of the rim was removed to prevent excessive UV absorption and light refraction from the plastic. A screen made from laminated printing paper was glued to it. Larvae were mounted in 2% low melting agarose (Sigma-Aldrich) and positioned in the center of the arena at ~7 mm from the screen. Once the agarose was set, the arena was filled with E3 medium and a scalpel was used to free both tail and head, making sure that the larva could move its eyes. The used tracking protocol measured eye-specific angle over time coupled with tail curvature. Both parameters were collected from the analysis, but tail tracking was not considered during data processing due to the very noisy traces given by the system's lightning conditions.

## Maximum intensity hunting assay

Larvae were exposed to a series of 20 stimuli proceeding in alternate directions. After 2 s of standby, the LED turns on, projecting a ~4.9° stimulus in front of the larva. The projected dot moves towards either left or right at ~33°/s, ending after 3 s at ~90° of the larval field of view, due to the properties of the arena. When the stimulus stops moving, it turns off the LED and resets its position. Stimuli alternate in direction and always start with a left-right movement in the larva perspective. All stimuli are delivered with LED intensity at 100%.

## Incremental intensity hunting assay

The assay is structured as described under the 'Maximum intensity hunting assay' section. Here, light intensity is delivered in 10 steps of intensity, with each step replicated for each stimulus direction (left-right, right-left). The intensity steps go from 0% (used as a no-stimulus control), up to 90%. Percentages reflect the analog value provided through the setup and not the actual intensity unit. We stopped at this level after observing a performance drop directly after intensity at 70% (probably due to stimulus habituation). Optical power analysis were performed to measure light intensity of each step with the different light sources. A Usb2000+ Spectrometer (OceanOptics) was used to record the stimuli absolute irradiance at the position of the paper screen. Due to the low intensity of the stimuli, irradiance traced resulted in very noisy. We nevertheless calculated the optical power (in µW) by integrating the absolute irradiance recorded from 300 to 800 nm and multiplying the value by the spectrometer sensor area. Measurements are shown in *Table 2*.

## Hunting assays data analysis

Raw tracking data obtained by Stytra (*Štih et al., 2019*) was analyzed in Python. The designed event-detection algorithm compares the raw eye angle tracks with a rolling average of the same trace to detect peaks and compare their amplitude with a set threshold derived from the average noise of the

**Table 2.** Optical power of the hunting assay light stimuli.

| Stimulus percentage | Orange – Optical power (µW) | UV – Optical power (µW) |
|---|---|---|
| 10% | 1.160 | 1.023 |
| 20% | 1.156 | 1.058 |
| 30% | 1.168 | 1.075 |
| 40% | 1.178 | 1.085 |
| 50% | 1.188 | 1.102 |
| 60% | 1.188 | 1.107 |
| 70% | 1.192 | 1.135 |
| 80% | 1.139 | 1.141 |
| 90% | 1.199 | 1.158 |

**Table 3.** Light stimuli and recording settings of the OMR assay.

| Red band digital contrasts and measured optical power (μW) | 170 (2.102 μW) – 43 (0.7643 μW) – 255 (2.3 μW – 85) (1.012 μW) – 0 (0.664 μW) – 213 (2.347 μW) – 128 (1.465 μW) |
|---|---|
| Red gratings wavelength range | >550 nm; peak at 596 nm |
| Gratings speed | Centering/Rightward: 14.5 mm/s<br>Habituation: 0 mm/s |
| Gratings period | 6 mm |
| Paradigm structure | Centering stimulus (30 s) – habituation (30 s) – centering stimulus (30 s) – rightward stimulus (30 s) |
| Camera exposure | 30 ms |

trace itself to adapt to potentially noisy recordings. The threshold value to surpass in order to count an event has been tweaked manually by comparison with operator-observed events. If both eyes have a convergence (inward turn of the eye) event (prey locking) or if one single eye strongly converges (prey tracking) over the set thresholds, an event is registered. Both prey locking and tracking events have been pooled for these experiments. The same procedures apply for the tail by comparing raw total tail curvature to a rolling averaged trace, assigning an event when the curvature peaks in the direction of the stimulus movement (tail recordings have not been used due to tracking difficulties). Intensity-specific comparisons between mutants and WT animals were performed in the form of chi-square tests of stimulus detections over stimuli display, but none proved to be significant ($p$-value>0.05) and were not included in the paper. A second script pools information from all the fish and plots fish-specific single events along with total event frequency per epoch or for each delivered stimulus. Total responses over delivered stimuli are tested for significance via two proportions z-test (CI measured via Wilson method), while intensity-specific response distributions are tested via Kolmogorov-Smirnov test.

## Long-wavelength specific OMR assay

OMR assays were performed on a custom-built setup composed of: a 3D printed arena (11 rows 1×16 cm) glued on clear glass and positioned on a diffuser panel, a projector (hp vp6111) to display moving gratings controlled by the Python-based software Stytra (*Štih et al., 2019*) and a high-speed camera (Stingray F046B ASG, Allied Vision) mounting an IR filter on a 12.5 mm objective (FUJINON CF12.5HA-1). A bottom IR light source (infrared spotlight baustein, Kemo Electronics) provides the illumination for the camera feed. The stimulation paradigm was divided into four stages: a centering stimulus with black and white gratings moving towards the center of the arena, a still red and black grating to observe basal movement direction preference, a second centering stimulus and finally a rightward moving grating stimulus. Zebrafish larvae were exposed to seven different red contrasts in a pseudo-random order. Specifics of the stimuli are showed in *Table 3*. Optical power analysis were performed to measure light intensity of each step with the different light sources. A Usb2000+Spectrometer (OceanOptics) was used to record the stimuli absolute irradiance at the position of the paper screen. Due to the low intensity of the stimuli, irradiance traced resulted in very noisy. We nevertheless calculated the optical power (in μW) by integrating the absolute irradiance recorded from 300 to 800 nm and multiplying the value by the spectrometer sensor area. Measurements are shown in *Table 3*. Raw tracking data (fish coordinates over time) was analyzed in Python. The recorded area from the camera was divided into 10 ROIs that included the 10 rows with larvae in it. Coordinates inside each ROI were assigned to the respective fish. Zebrafish larvae travelled distance was measured by calculating the distance between consecutive coordinates and summing them. Movement towards the stimulus direction (rightward) was measured by filtering through consecutive coordinates for which the x values were increasing. Speed between each coordinate was calculated and filtered for movements over double their average speed (~29 mm/s) (*Clift et al., 2014*). Data from multiple assays was pooled and rightward movement during both habituation (still colored grating) and stimulus phase (rightward-moving colored grating) were compared. Statistical significance was calculated by repeated measures ANOVA with the software GraphPad Prism 9.5.0.

## Acknowledgements

All authors would like to thank Martin Walther, Kara Kristiansen, and Heidi Mockel for excellent technical assistance and animal care taking. We would also like to thank Prof. Dr. Tom Baden and Prof. Dr. Martin Müller for their excellent suggestions and insights throughout the whole project. This work was funded by the Swiss National Science Foundation (310030_204648 and 310030_200376).

## Additional information

### Funding

| Funder | Grant reference number | Author |
|---|---|---|
| Schweizerischer Nationalfonds zur Förderung der Wissenschaftlichen Forschung | 310030_204648 | Stephan CF Neuhauss |
| Schweizerischer Nationalfonds zur Förderung der Wissenschaftlichen Forschung | 310030_135598 | Stephan CF Neuhauss |

The funders had no role in study design, data collection and interpretation, or the decision to submit the work for publication.

### Author contributions

Marco Garbelli, Conceptualization, Resources, Data curation, Software, Formal analysis, Validation, Investigation, Visualization, Methodology, Writing – original draft; Stephanie Niklaus, Resources, Investigation; Stephan CF Neuhauss, Conceptualization, Resources, Supervision, Funding acquisition, Project administration, Writing – review and editing

### Author ORCIDs

Marco Garbelli  https://orcid.org/0000-0001-8782-5934
Stephan CF Neuhauss  https://orcid.org/0000-0002-9615-480X

### Ethics

All animal experiments were non-invasive behavioral tests. They were carried out in line with the ARVO Statement for the Use of Animals in Ophthalmic and Vision Research and were approved by the Veterinary Authorities of Kanton Zurich, Switzerland (TV4206, ZH064/2024, ZH032/18).

Reviewer #1 (Public review): https://doi.org/10.7554/eLife.102346.3.sa1
Reviewer #2 (Public review): https://doi.org/10.7554/eLife.102346.3.sa2
Author response https://doi.org/10.7554/eLife.102346.3.sa3

## Additional files

### Supplementary files
MDAR checklist

### Data availability
All behavioral, immunohistochemical, and electrophysiological raw data and code is provided on Dryad (https://doi.org/10.5061/dryad.gf1vhhn2x).

The following dataset was generated:

| Author(s) | Year | Dataset title | Dataset URL | Database and Identifier |
|---|---|---|---|---|
| Garbelli M, Niklaus S, Neuhauss SCF | 2025 | Characterization of Postsynaptic Glutamate Transporter Functionality in the Zebrafish Retinal First Synapse Across Different Wavelengths | https://doi.org/10.5061/dryad.gf1vhhn2x | Dryad Digital Repository, 10.5061/dryad.gf1vhhn2x |

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
