## [Editor Report · eLife Assessment]

This **important** study reveals that Excitatory Amino Acid Transporters play a role in chromatic information processing in the retina. The combination of (double) mutants, behavioral assays, immunohistochemistry, and electroretinograms provides **solid** evidence supporting the appropriately conservative conclusions. The work will be of interest to neurobiologists working on color vision or retinal processing.

---

## [Referee Report · Reviewer #1 (Public review)]

Summary:

This manuscript by Garbelli et al. investigates the roles of excitatory amino acid transporters (EAATs) in retinal bipolar cells. The group previously identified that EAAT5b and EAAT7 are expressed at the dendritic tips of bipolar cells, where they connect with photoreceptor terminals. The previous study found that the light responses of bipolar cells, measured by electroretinogram (ERG) in response to white light, were reduced in double mutants, though there was little to no reduction in light responses in single mutants of either EAAT5b or EAAT7.

The current study further explores the roles of EAAT5b and EAAT7 in bipolar cells' chromatic responses. The authors found that bipolar cell responses to red light, but not to green or UV-blue light, were reduced in single mutants of both EAAT5b and EAAT7. In contrast, UV-blue light responses were reduced in double mutants. Additionally, the authors observed that EAAT5b, but not EAAT7, is strongly localized in the UV cone-enriched area of the eye, known as the "Strike Zone (SZ)." This led them to investigate the impact of the EAAT5b mutation on prey detection performance, which is mediated by UV cones in the SZ. Surprisingly, contrary to the predicted role of EAAT5b in prey detection, EAAT5b mutants did not show any changes in prey detection performance compared to wild-type fish. Interestingly, EAAT7 mutants exhibited enhanced prey detection performance, though the underlying mechanisms remain unclear.

The distribution of EAAT7 protein in the outer plexiform layer across the eye correlates with the distribution of red cones. Based on this, the authors tested the behavioral performance driven by red light in EAAT5b and EAAT7 mutants. The results here were again somewhat contrary to predictions based on ERG findings and protein localization: the optomotor response was reduced in EAAT5b mutants, but not in EAAT7 mutants.

Strengths:

Although the paper lacks cohesive conclusions, as many results contradict initial predictions as mentioned above, the authors discuss possible mechanisms for these contradictions and suggest future avenues for study. Nevertheless, this paper demonstrates a novel mechanism underlying chromatic information processing.

The manuscript is well-written, the data are well-presented, and the analysis is thorough.

Weaknesses:

I have only a minor comment. The authors present preliminary data on mGluR6b distribution across the eye. Since this result is based on a single fish, I recommend either adding more samples or removing this data, as it does not significantly impact the paper's main conclusions.

Comments on revisions:

The authors addressed all of the concerns that I had in the original manuscript.

---

## [Referee Report · Reviewer #2 (Public review)]

Garbelli et. al. set out to elucidate the function of two glutamate transporters, EAAT5b and EAAT7, in the functional and behavioral responses to different wavelengths of light. The question is an interesting one because these transporters are well-positioned to affect responses to light, and their distribution in the retina suggests that they could play differential roles in visual behaviors. However, the resolution of the functional and behavioral data presented here means that the conclusions are necessarily a bit vague.

In Figure 1, the authors show that the double KO has a decreased ERG response to UV/blue and red wavelengths. However, the individual mutations both only affect the response to red light, suggesting that they might affect behaviors such as OMR that typically rely on this part of the visual spectrum. However, there was no significant change in the response to UV/blue light of any intensity, making it unclear whether the mutations could individually play roles in detection of UV prey. Based on the later behavioral data, it seems likely that at least the EAAT7 KO should affect retinal responses to UV light, but it may be that the ERG does not have the spatial or temporal resolution to detect the difference, or that the presence of blue light overwhelmed any effect of the individual knockouts on the response to UV light.

In Figures 5 and 6, the authors compare the two knockouts to wild-type fish in terms of their sensitivity to UV prey in a hunting assay. The EAAT5b KO showed no significant impairment in UV sensitivity, while the EAAT7 KO fish actually had an increased hunting response to UV prey. However, there is no comparison of the KO and WT responses to different UV intensities, only in bulk, so we cannot conclude that the EAAT7 KO is allowing the fish to detect weaker prey-like stimuli.

In Figure 7, the EAAT5b KO seems to cause a decrease in OMR behavior to red grating stimuli, but only one stimulus is tested, so it is unclear whether this is due to a change in visual sensitivity or resolution.

The conclusions made in the manuscript are appropriately conservative; the abstract states that these transporters somehow influence prey detection and motion sensing, and this is likely true.

In terms of impact on the field, this work highlights the potential importance of these two transporters to visual processing, but further studies will be required to say how important they are and exactly what they are doing.

---

## [Author Response]

The following is the authors’ response to the original reviews

**Public Reviews:**

**Reviewer #1 (Public review):**
Summary:This manuscript by Garbelli et al. investigates the roles of excitatory amino acid transporters (EAATs) in retinal bipolar cells. The group previously identified that EAAT5b and EAAT7 are expressed at the dendritic tips of bipolar cells, where they connect with photoreceptor terminals. The previous study found that the light responses of bipolar cells, measured by electroretinogram (ERG) in response to white light, were reduced in double mutants, though there was little to no reduction in light responses in single mutants of either EAAT5b or EAAT7.The current study further explores the roles of EAAT5b and EAAT7 in bipolar cells' chromatic responses. The authors found that bipolar cell responses to red light, but not to green or UV-blue light, were reduced in single mutants of both EAAT5b and EAAT7. In contrast, UV-blue light responses were reduced in double mutants. Additionally, the authors observed that EAAT5b, but not EAAT7, is strongly localized in the UV cone-enriched area of the eye, known as the "Strike Zone (SZ)." This led them to investigate the impact of the EAAT5b mutation on prey detection performance, which is mediated by UV cones in the SZ. Surprisingly, contrary to the predicted role of EAAT5b in prey detection, EAAT5b mutants did not show any changes in prey detection performance compared to wild-type fish. Interestingly, EAAT7 mutants exhibited enhanced prey detection performance, though the underlying mechanisms remain unclear.The distribution of EAAT7 protein in the outer plexiform layer across the eye correlates with the distribution of red cones. Based on this, the authors tested the behavioral performance driven by red light in EAAT5b and EAAT7 mutants. The results here were again somewhat contrary to predictions based on ERG findings and protein localization: the optomotor response was reduced in EAAT5b mutants, but not in EAAT7 mutants.Strengths:Although the paper lacks cohesive conclusions, as many results contradict initial predictions as mentioned above, the authors discuss possible mechanisms for these contradictions and suggest future avenues for study. Nevertheless, this paper demonstrates a novel mechanism underlying chromatic information processing.The manuscript is well-written, the data are well-presented, and the analysis is thorough.

We are happy about the perceived strengths of our manuscript.

Weaknesses:I have only a minor comment. The authors present preliminary data on mGluR6b distribution across the eye. Since this result is based on a single fish, I recommend either adding more samples or removing this data, as it does not significantly impact the paper's main conclusions.

We agree that the mGluR6 result is statistically underpower (we would never claim differently). The data is based on only one clutch of fish, comprising 11 eyes. Since the data is anyway in the supplement and not part of the main story, we would like to keep it to spur further investigations into anisotropic distribution of synaptic proteins.

**Reviewer #2 (Public review):**
Garbelli et. al. set out to elucidate the function of two glutamate transporters, EAAT5b and EAAT7, in the functional and behavioral responses to different wavelengths of light. The question is an interesting one, because these transporters are well positioned to affect responses to light, and their distribution in the retina suggests that they could play differential roles in visual behaviors. However, the low resolution of both the functional and behavioral data presented here means that the conclusions are necessarily a bit vague.In Figure 1, the authors show that the double KO has a decreased ERG response to UV/blue and red wavelengths. However, the individual mutations only affect the response to red light, suggesting that they might affect behaviors such as OMR which typically rely on this part of the visual spectrum. However, there was no significant change in the response to UV/blue light of any intensity, making it unclear whether the mutations could individually play roles in the detection of UV prey. Based on the later behavioral data, it seems likely that at least the EAAT7 KO should affect retinal responses to UV light, but it may be that the ERG does not have the spatial or temporal resolution to detect the difference, or that the presence of blue light overwhelmed any effect of the individual knockouts on the response to UV light.In Figures 5 and 6, the authors compare the two knockouts to wild-type fish in terms of their sensitivity to UV prey in a hunting assay. The EAAT5b KO showed no significant impairment in UV sensitivity, while the EAAT7 KO fish actually had an increased hunting response to UV prey. However, there is no comparison of the KO and WT responses to different UV intensities, only in bulk, so we cannot conclude that the EAAT7 KO is allowing the fish to detect weaker prey-like stimuli.

We have now reported in both in the results paragraph and in the methods section that response-comparison of intensity-specific responses were non-significant in all instances of analyses (Chi-square test with p>0.05). We decided not to add the information to the figure as it does not add to the data and risks causing excessive clutter of an already complex graph.

As reviewer #2 rightfully states, we cannot conclude that EAAT7 KO is allowing the fish to detect weaker prey-like stimuli. We only intend to suggest that a lack of EAAT7 might facilitate prey detection events as the number of hunting events in total, is increased compared to WT.

In Figure 7, the EAAT5b KO seems to cause a decrease in OMR behavior to red grating stimuli, but only one stimulus is tested, so it is unclear whether this is due to a change in visual sensitivity or resolution.

We fully agree that further experiments presenting different stimuli in the setup may very well reveal more details on the nature of the observed defect and thank reviewer #2 for the suggestion. We feel that identifying the reason of the defect lies outside of the scope of this paper, but should definitely be investigated in future studies.

The conclusions made in the manuscript are appropriately conservative; the abstract states that these transporters somehow influence prey detection and motion sensing, and this is probably true. However, it is unclear to what extent and how they might be acting on these processes, so the conclusions are a bit unsatisfying.In terms of impact on the field, this work highlights the potential importance of these two transporters to visual processing, but further studies will be required to say how important they are and what they are doing. The methods presented here are not novel, as UV prey and red OMR stimuli and behaviors have previously been described.

We agree that this study is not fully conclusive but a first step towards a clarification of the role of glutamate transporters in shaping visual behavior.

**Recommendations for the authors:**

**Reviewer #2 (Recommendations for the authors):**
Suggestions for improved or additional experiments, data, or analyses:Figure 3:(a) What is the intensity of the light emitted by the UV and yellow LEDs and experienced by the larva, e.g. in nW? This is necessary in order to be able to compare and replicate the results.

Stimuli intensities in microwatts are now included and reported in the Materials and Methods sections

(b) In Figure 3D, are all the example eye movement events hunting initiations? Does right eye/left eye positive or negative angle change denote convergence?

As indicated in the figure legend, hunting initiations are indicated by black dots on the graph. In Stytra’s eye tracking system, eye convergence is indicated by an increase in the left eye angle and a decrease in the right eye angle. Both these points have now been clarified in the figure legend.

(c) Also in 3D, the tail angle plot and x-axis are too small to read.

Figure 3D has been reformatted to be more legible.

(d) How much eye convergence constitutes a response? In order to compare the findings to previous studies of prey capture, it would be best to use a bimodal distribution of eye angles to set a convergence threshold for each fish (e.g. Paride et. al., eLife 2019), but there should at least be a clear threshold mentioned.

We have expanded the explanation of how the response detection paradigm was calculated. We acknowledge that this analysis has limitations in terms of comparability with previous studies, as it was developed de novo, based on the format of eye coordinate data provided by Stytra and refined through iterative comparison with experimental video recordings. Since the threshold was defined relative to the average noise level of the trace, it is difficult to specify an exact value. However, we are happy to share the Python scripts used for the analysis to facilitate further investigation.

(e) The previous study using artificial UV prey stimuli to trigger hunting (Khan et. al., Current Biology 2023) should be acknowledged.

This is an indeed an embarrassing omission, not excused by the first version of this section being drafted before the Khan publication. We have now cited this important study.

Figure 5:Was the response at any individual intensity significantly lower in the mutant? If not, this should be clearly stated.

Yes, and this is now clearly stated in the main text

Figure 6:Again, it would be more informative to know for which intensities the KO response was significantly greater than WT.

This is now also clearly stated in the main text

Figure 7:(a) What are the intensity units?

We now clarified in the figure that the intensity shown in the graph is digital intensity

(b) Similar to Figures 5 and 6, it would be more informative to know at which intensities the KO response was significantly different from WT.

We now report the measured optical powers relative to the digital intensities in the Materials and Methods sections.

Suggestion for writing:The discussion was a bit discursive. A more structured discussion, sequentially explaining each of the key results, would be easier for the reader to follow. And, it would be helpful to have hypotheses for how these transporter mutants could cause each of the changes in visual behaviors that were observed.

We agree that the discussion needed improvements. We have completely rewritten the discussion and hope that it now more concisely put our results into context.